
# On the descriptive power of Neural Networks as constrained Tensor Networks with exponentially large bond dimension

**Mario Collura[1,2,3], Luca Dell'Anna[2,4], Timo Felser[1,2,5] and Simone Montangero[2,4,5]**

**1** Theoretische Physik, Universität des Saarlandes, D-66123 Saarbrücken, Germany.
**2** Dipartimento di Fisica e Astronomia, "G. Galilei",
Università di Padova, I-35131 Padova, Italy.
**3** SISSA – International School for Advanced Studies, I-34136 Trieste, Italy.
**4** Padua Quantum Technologies Research Center, Università di Padova, I-35131 Padova, Italy.
**5** INFN, Sezione di Padova, Via Marzolo 8, I-35131 Padova, Italy.

## Abstract

In many cases, neural networks can be mapped into tensor networks with an exponentially large bond dimension. Here, we compare different sub-classes of neural network states, with their mapped tensor network counterpart for studying the ground state of short-range Hamiltonians. We show that when mapping a neural network, the resulting tensor network is highly constrained and thus the neural network states do in general not deliver the naive expected drastic improvement against the state-of-the-art tensor network methods. We explicitly show this result in two paradigmatic examples, the 1D ferromagnetic Ising model and the 2D antiferromagnetic Heisenberg model, addressing the lack of a detailed comparison of the expressiveness of these increasingly popular, variational ansätze.

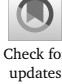
# 1   Introduction

Artificial Neural Networks (NN) have increasingly taken hold in various research fields and technology [1–3]. Their power in recognizing special patterns behind a huge amount of raw data allowed a revolutionary change in our approach towards deep learning [4, 5]. Taking inspiration from biological neural networks, NNs can be a good framework to process big sets of data. As a matter of fact, NNs can be seen as special functional mappings of many variables (physical and hidden), which can be trained by specific algorithms and applied to a very broad spectrum of applications in different fields, one of which is statistical physics [2,6–9]. A powerful example is the Restricted Boltzmann Machine (RBM) which has been largely employed to mimic the behaviour of complex quantum systems [10–17]. Essentially, RBM is a type of artificial neural network which, in interacting quantum systems, can be understood as a particular variational ansatz for the many-body wave function.

Another very successful class of wave-function variational ansatz that has been widely exploited are Tensor Network (TN) states [18–34]. They are based on the replacement of the non-local rank-$N$ tensor representing the $N$-body wave function, with $O(N)$ local tensors with smaller rank, connected via auxiliary indexes. Such ansatz interpolates between the mean-field approach, where quantum correlations are completely neglected, and the exact (but inefficient) representation of the state. The interpolation is governed by the dimension $\chi$ of the auxiliary indexes connecting local tensors. In one dimension, a very successful tensor network representation is the so-called Matrix Product States (MPS) [18, 19, 25, 33].

When applied to the study of many-body quantum systems, these two very powerful approaches reduce the exponentially large Hilbert space dimension by optimally tuning a number of parameters which scales polynomially with $N$. In particular, the number of free parameters scales as $O(NM)$ for a fully connected RBM (where $M$ is the number of hidden variables), while is $O(N\chi^2)$ for an MPS. Recently, a strong connection between NN and TN has been pointed out [40–46]: among others, it has been shown that the fully connected RBM can be explicitly rewritten as a MPS with an exponentially large auxiliary dimension, i.e. $\chi = 2^M$ (see Fig. 1). Considering the fact that the bipartite entanglement entropy in an MPS is proportional to $\log \chi$, suggests that RBMs may provide a way to represent highly correlated quantum states whose entanglement content scales with the system volume [47], thus going much beyond the MPS descriptive power that is limited to area-law states [48, 49].

Here, present a systematic comparison between constraint TN representations of NNs and the unconstrained counterparts aiming to investigating the actual descriptive power of the NN ansatz as constrained Tensor Networks with exponentially large bond dimension. With this comparison we further aim to address the lack of a detailed comparison of the expressiveness of these various ansatz considering the increasing popularity of such variational states and encourage further work in this direction. In the following section, we unveil fundamental aspects of this important connection between TN and NN states and along the way we introduce a new mapping between NN and TN, valid also in two-dimensions. This mapping can be exploited to introduce more efficient strategies to optimize NN states. Finally, we present two paradigmatic examples, the study of one-dimensional and two-dimensional ground states of many-body quantum Hamiltonians and show that a TN with moderate bond dimensions can match – if not overcome – the prediction of the correspondent NN, that were expected to deliver results equivalent to those obtained via TN with an in practice unreachable bond dimension. In particular, we compare the prediction of NN and TN, and show that the latter are able to deliver higher precision in local quantities and in correlation functions with respect to the equivalent NN. We further clarify why these results have to be expected.

In one dimension, these results are based on the fact that beyond the naive expectations, the detailed aspects in the relationship between RBM and MPS shall be considered. In partic-

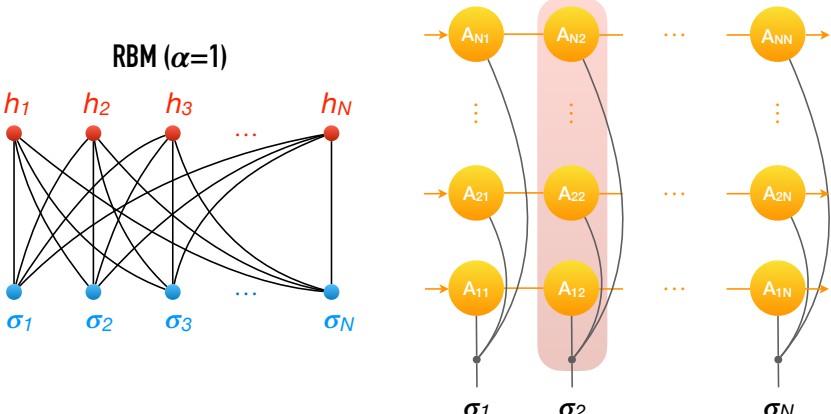

Figure 1: The RBM for $\alpha = 1$ ($M = N$) (left) can be mapped to an exponentially large coMPS (right). The pink-shaded region represents the $2^N \times 2^N$ local matrix $\mathbf{\Sigma}_j^{\sigma_j}$ which is constructed as a tensor product of $N$ matrices (yellow circles) with bond dimension $\chi = 2$ (see main text for details). Arrows indicate periodic boundary conditions.

ular, *(i)* different topologies of the NN may result in MPSs with very different auxiliary dimensions; *(ii)* the emerging "constrained" MPS (coMPS) is highly constrained due to the mapping itself. In practice, even though the formal mapping reveals an exponentially large auxiliary dimension of the related coMPS, the number of independent entries of the tensors scales polynomially with the size of the artificial neural network. This makes the coMPS representation of the RBM inefficient.

The first point above raises a very delicate question regarding the efficiency of the NN *tout court*. Physical many-body states are usually eigenstates of short-range interacting Hamiltonians, which naturally introduce the notion of distance between lattice sites, suggesting that a valuable approach should encode this information *ab initio*. Therefore, a useful modification of the RBM is obtained by introducing proper intra-layer connections in the wave function ansatz; the resulting unRestricted Boltzmann Machine (uRBM) has been recently employed in order to describe the ground state of the ferromagnetic Ising quantum chain [50]. A simple structure with one layer of hidden variables (see Fig. 2) explicitly encodes the underlying Hamiltonian geometry and makes it possible to obtain an increased accuracy with respect to the corresponding RBM (i.e. with $\alpha = 1$), with a much smaller number of free parameters. The surprising effect of this result will become evident in the following, when the RBM and the uRBM will be compared at the level of the mapping to the corresponding coMPS.

## 2 Constrained Matrix Product States

Here we introduce the "constrained" Matrix Product State, highlighting the differences between RBM and uRBM. In the following, periodic boundary conditions are assumed, and results are valid for $N > 2$.

RBM: An RBM is defined as follows: given a set $\boldsymbol{\sigma} = \{\sigma_1, \sigma_2, \ldots, \sigma_N\}$ of $N$ physical binary variables (e.g. the eigenvalues $\sigma_j = \pm 1$ of a spin-1/2), one introduces an extra set of "unphysical" hidden variables $\boldsymbol{h} = \{h_1, h_2, \ldots, h_M\}$ (with density $\alpha \equiv M/N$) such that the unnormalised many-body wave function of the RBM type is obtained from a full Boltzmann distribution by

tracing out the hidden set,

$$\Psi_\alpha(\boldsymbol{\sigma}) = \sum_{\boldsymbol{h}} \exp\left[-\psi_\alpha(\boldsymbol{\sigma}, \boldsymbol{h})\right], \tag{1}$$

where $\psi_\alpha$ is the RBM's functional

$$\psi_\alpha(\boldsymbol{\sigma}, \boldsymbol{h}) = \sum_{j=1}^{N} a_j \sigma_j + \sum_{j=1}^{M} b_j h_j + \sum_{i,j=1}^{M,N} \Gamma_{ij} h_i \sigma_j, \tag{2}$$

and $\alpha = M/N$ is assumed to be a positive integer for convenience. Parameters $a_j$ and $b_j$ are local biases (in the spin-1/2 picture, they represent local magnetic fields) applied to the variables, whilst $\Gamma_{ij}$ are couplings between the physical and the hidden variables (see Fig. 1).

As stated in the introduction, in the previous prescription there is no direct connections within the same set of variables while the two sets are fully connected, thus there is no notion of a distance. Nonetheless, correlations between physical variables may be mediated by their fictitious interactions with the hidden variables. A priori the RBM variational ansatz is well suited to work in any dimension. As a matter of fact, it is a promising tool to describe many-body quantum states [35–39]; recently convolutional NNs have been employed to improve the level of accuracy of the shallow RBMs in order to deal with frustrated 2D lattice models [51].

RBM wave functions can be rewritten as a coMPS with periodic boundary conditions. The absence of intra-layer couplings allows to easily take the sum over the hidden variables in Eq. (1), thus obtaining [40]

$$\Psi_\alpha(\boldsymbol{\sigma}) = e^{-\sum_{j=1}^{N} a_j \sigma_j} \prod_{i=1}^{M} 2\cosh\left(b_i + \sum_{j=1}^{N} \Gamma_{ij} \sigma_j\right) = \mathrm{Tr}\left[\prod_{j=1}^{N} \Sigma_j^{\sigma_j}\right] \tag{3}$$

in terms of $2^M \times 2^M$ real diagonal matrices (see Fig. 1) of the form

$$\Sigma_j^\sigma = e^{-a_j \sigma} \bigotimes_{i=1}^{M} \begin{pmatrix} e^{-b_i/N - \Gamma_{ij}\sigma} & 0 \\ 0 & e^{b_i/N + \Gamma_{ij}\sigma} \end{pmatrix}. \tag{4}$$

Notice that, if the RBM wave function describes a translational invariant quantum state, we should have $\Psi_\alpha(\boldsymbol{\sigma}) = \Psi_\alpha(\boldsymbol{\sigma}')$, where $\boldsymbol{\sigma}$ and $\boldsymbol{\sigma}'$ differ for an arbitrary cyclic permutation of local spin variables: thus, all local tensors $\Sigma_j^{\sigma_j}$ can be set to be equal and independent of the lattice site $j$, reducing the number of free parameters to $2M + 1$. Let us mention that when the local biases are set to zero, the wave function becomes spin-flip invariant as well.

1D - uRBM: Let us now turn our attention to the unrestricted Boltzmann machine for 1D systems. When explicitly encoding the geometry of the underlying model into the artificial neural network, we can describe the many-body quantum state via the following uRBM un-normalised wave function

$$\Phi_\ell(\boldsymbol{\sigma}) = \sum_{\boldsymbol{h}} \exp\left[-\phi_\ell(\boldsymbol{\sigma}, \boldsymbol{h})\right], \tag{5}$$

where now the hidden variables are labeled according to $\boldsymbol{h} = \{h_j^\gamma\}$ with $j \in [1, N]$, and $\gamma \in [1, \ell]$ denoting the different layers. The uRBM's functional is now given by

$$\begin{aligned} \phi_\ell(\boldsymbol{\sigma}, \boldsymbol{h}) &= \sum_{j=1}^{N}\Big[ K_j^0 \sigma_j \sigma_{j+1} + \sum_{\gamma=1}^{\ell} K_j^\gamma h_j^\gamma h_{j+1}^\gamma \\ &+ J_j^1 \sigma_j h_j^1 + \sum_{\gamma=2}^{\ell} J_j^\gamma h_j^{\gamma-1} h_j^\gamma \Big]. \end{aligned} \tag{6}$$

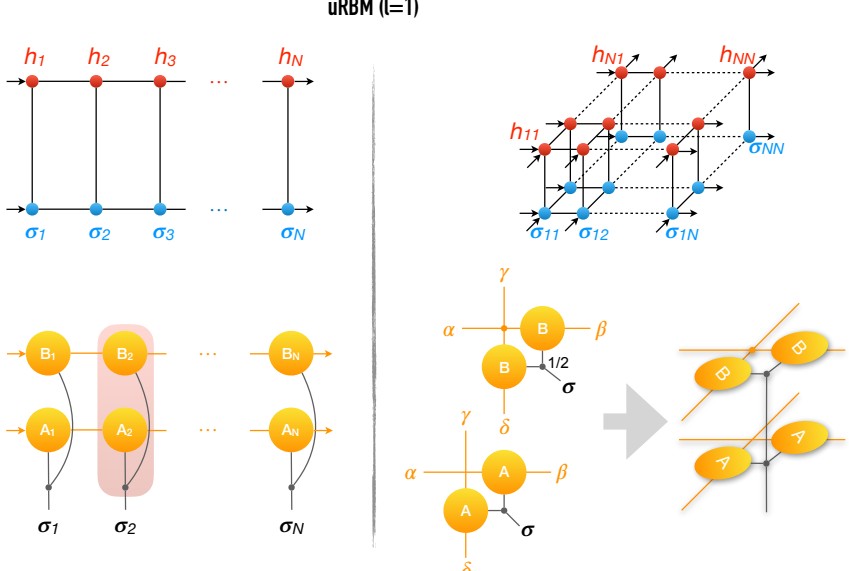

Figure 2: The uRBM with one layer of hidden variables in one (two) dimension is mapped to the corresponding coMPS (coPEPS). In the 2D case we only draw the local building block of the full tensor network. Here in general crossing lines are independent, except when a dot fixes them to be equals. The 1/2 in one of the dot means that the $B$ matrices have to be evaluated at $\sigma/2$ (see main text for details). Arrows indicate periodic boundary conditions.

Interestingly, the state described by the uRBM wave function enforces the spin-flip invariance of the Ising Hamiltonian, namely $\Phi_\ell(-\boldsymbol{\sigma}) = \Phi_\ell(\boldsymbol{\sigma})$; moreover, it is also invariant under the transformation $J_j^\gamma \rightarrow -J_j^\gamma$, for arbitrary $\gamma$. Although we cannot analytically sum over the hidden variables as before, it is still possible to trace-out the hidden variables recasting Eq. (5) in a simple coMPS form. Exploiting the transfer matrix approach for evaluating the partial partition function, we obtain (e.g. for $\ell = 1$)

$$\Phi_1(\boldsymbol{\sigma}) = \text{Tr}\left[\prod_{j=1}^N (A_j^{\sigma_j} \otimes B_j^{\sigma_j})\right], \tag{7}$$

where

$$A_j^\sigma = \begin{pmatrix} \cosh(K_j^0) & -\sinh(K_j^0)\sigma \\ \cosh(K_j^0)\sigma & -\sinh(K_j^0) \end{pmatrix}, \tag{8}$$

$$B_j^\sigma = \begin{pmatrix} e^{-K_j^1 - J_j^1\sigma} & e^{K_j^1 - J_j^1\sigma} \\ e^{K_j^1 + J_j^1\sigma} & e^{-K_j^1 + J_j^1\sigma} \end{pmatrix}. \tag{9}$$

Also, in this case, translational invariance of the many-body state can be exploited reducing the number of free parameters by a factor $N$. Interestingly, the spin-flip symmetry of the state is reflected in the coMPS representation as the local invariance $A_j^{-\sigma} = \hat\sigma^z A_j^\sigma \hat\sigma^z$, $B_j^{-\sigma} = \hat\sigma^x B_j^\sigma \hat\sigma^x$. Finally, the mapping can be extended to an arbitrary number of additional hidden layers which results in a coMPS with auxiliary dimension $\chi_\ell = 2^{\ell+1}$.

2D – uRBM: The geometry encoded in the uRBM may affect the tensor network representation of the many body wave function which, in the 2D cases, can be written as a coPEPS

(constrained Projected Entangled Pair State). For the sake of simplicity, we focus only on the translational invariant case where the 2D uRBM wave function reads

$$\Phi_\ell^{2D}(\boldsymbol{\sigma}) = \sum_{\boldsymbol{h}} \exp\left[-\phi_\ell^{2D}(\boldsymbol{\sigma}, \boldsymbol{h})\right], \tag{10}$$

with

$$
\begin{aligned}
\phi_\ell^{2D}(\boldsymbol{\sigma}, \boldsymbol{h}) \;=\; & \sum_{i,j=1}^{N} \Big[ K^0 (\sigma_{i,j}\sigma_{i,j+1} + \sigma_{i,j}\sigma_{i+1,j}) \\
& + \sum_{\gamma=1}^{\ell} K^\gamma (h_{i,j}^\gamma h_{i,j+1}^\gamma + h_{i,j}^\gamma h_{i+1,j}^\gamma) \\
& + J^1 \sigma_{i,j} h_{i,j}^1 + \sum_{\gamma=2}^{\ell} J^\gamma h_{i,j}^{\gamma-1} h_{i,j}^\gamma \Big].
\end{aligned}
\tag{11}
$$

Summing over the hidden variables, the wave function can be rewritten as a translational invariant coPEPS built from the local tensors (see Fig. 2)

$$
\begin{aligned}
\mathbb{A}_{\alpha\beta\gamma\delta}^\sigma \;&=\; (\boldsymbol{A}^\sigma)_{\alpha\beta}(\boldsymbol{A}^\sigma)_{\gamma\delta}, \\
\mathbb{B}_{\alpha'\beta'\gamma'\delta'}^\sigma \;&=\; \delta_{\alpha'\gamma'}(\boldsymbol{B}^{\sigma/2})_{\alpha'\beta'}(\boldsymbol{B}^{\sigma/2})_{\gamma'\delta'}.
\end{aligned}
\tag{12}
$$

with matrices $\boldsymbol{A}^\sigma$ and $\boldsymbol{B}^\sigma$ given by Eqs. (8) and (9), where we discarded the label $j$ due to the translational invariance. The local building block for the coPEPS is obtained by index fusion, paring each couple of indices to a single index which spans a four-dimensional auxiliary space, i.e. $\boldsymbol{\alpha} = (\alpha, \alpha')$, getting $\mathbb{C}_{\boldsymbol{\alpha\beta\gamma\delta}}^\sigma = \mathbb{A}_{\alpha\beta\gamma\delta}^\sigma \mathbb{B}_{\alpha'\beta'\gamma'\delta'}^\sigma$. Again, in this case, the extension to an arbitrary number of hidden layers is straightforward, and the coPEPS auxiliary dimension is the same as in the 1D case, namely $\chi_\ell = 2^{\ell+1}$.

The coMPS (coPEPS) mapping of the uRBM variational ansatz can be proficiently used together with Monte Carlo techniques in order to avoid the full sampling over the set of hidden variables. Indeed, those representations are a practical way to explicitly trace out the full set of the hidden variables.

## 3 Numerical results

In what follows we investigate how the different ansätze are able to describe the ground state of critical Hamiltonians (both in 1D and 2D), where bipartite entanglement entropy scales logarithmically with the system size, and the correlation functions decay algebraically.

### 3.1 The Ising quantum chain

We start our analysis with the ferromagnetic Ising quantum chain, whose Hamiltonian, for $N$ lattice sites and periodic boundary conditions, is given by

$$H_I = -\sum_{j=1}^{N} \hat{\sigma}_j^z \hat{\sigma}_{j+1}^z - \lambda \sum_{j=1}^{N} \hat{\sigma}_j^x, \tag{13}$$

where $\hat{\sigma}_j^\gamma$ (for $\gamma \in \{x, y, z\}$) are Pauli matrices acting on the site $j$, and $\hat{\sigma}_{N+1}^\gamma = \hat{\sigma}_1^\gamma$. The transverse field $\lambda$ drives the ground state from a ferromagnetic region ($\lambda < 1$) to a paramagnetic region ($\lambda > 1$) across a quantum critical point.

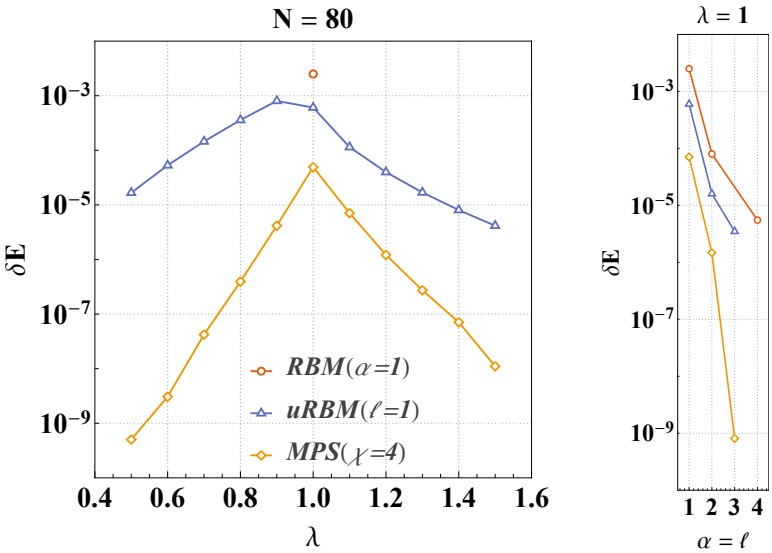

Figure 3: Relative error in the ground-state energy estimate for different many-body wave function representations. (left panel) We compare the uRBM with $\ell = 1$ with respect to the canonical MPS with the same bond dimension ($\chi = 2^{\ell+1} = 4$) as a function of the transverse field $\lambda$. (right panel) Scaling analysis of the energy error at the critical point, as a function of the hidden variable density $\alpha = \ell$ (for RBM and uRBM) and analogous bond dimension $\chi = 2^{\ell+1}$ (for the MPS). RBM (uRBM) representation reaches (overtakes) the accuracy of the MPS with $\chi = 4$ only for $\alpha = 2$ ($\ell = 2$); however, they remain above the estimate obtained with a canonical MPS with the same auxiliary dimension of the uRBM, i.e. $\chi = 2^{\ell+1} = 8$.

Exploiting the coMPS mapping of the uRBM, we are able to optimize the many-body wave function very efficiently. We consider a chain with periodic boundary conditions and mainly focus on the one layer case ($\ell = 1$), thus reducing the number of variational parameters to 3. Due to the coMPS representation of the variational wave function in Eq. (7) we are able to evaluate the Hamiltonian expectation value exactly. Thus, we improve the accuracy and the computational time compared to what has been recently found for the ground state energy with an uRBM in Ref. [50] via Monte Carlo methods. For sake of clarity, we point out that this approach drastically improves the evaluation of expectation values only and does not affect the time required for the optimisation of the uRBM wave-function.

In the left panel of Fig. 3 we report the relative error of the best estimate of the ground-state energy with respect to the exact value, namely $\delta E = |(\langle H_I \rangle - E_{ex})/E_{ex}|$, for a system of size $N = 80$ and varying the transverse field $\lambda \in [0.5, 1.5]$. We compare the results of the uRBM with $\ell = 1$ against the data obtained with a traditional MPS-based algorithm [52] with the same auxiliary dimension $\chi = 2^{\ell+1} = 4$. At the critical point, we also report the result obtained in Ref. [13] with the RBM variational ansatz and the *same number of hidden variables* (i.e. $\alpha = 1$). We confirm that appropriate physical insights about the model under investigation not only reduce the computational effort of the algorithm (from $2N + 1$ parameters in the RBM to 3 parameters in the uRBM), but results in higher precision. However, we notice that results based on the canonical MPS representation are order of magnitudes more accurate than those based on the corresponding uRBM representation.

Of course, it must be said that the canonical MPS representation contains more variational parameters than the uRBM with the same bond dimension; we would expect all represen-

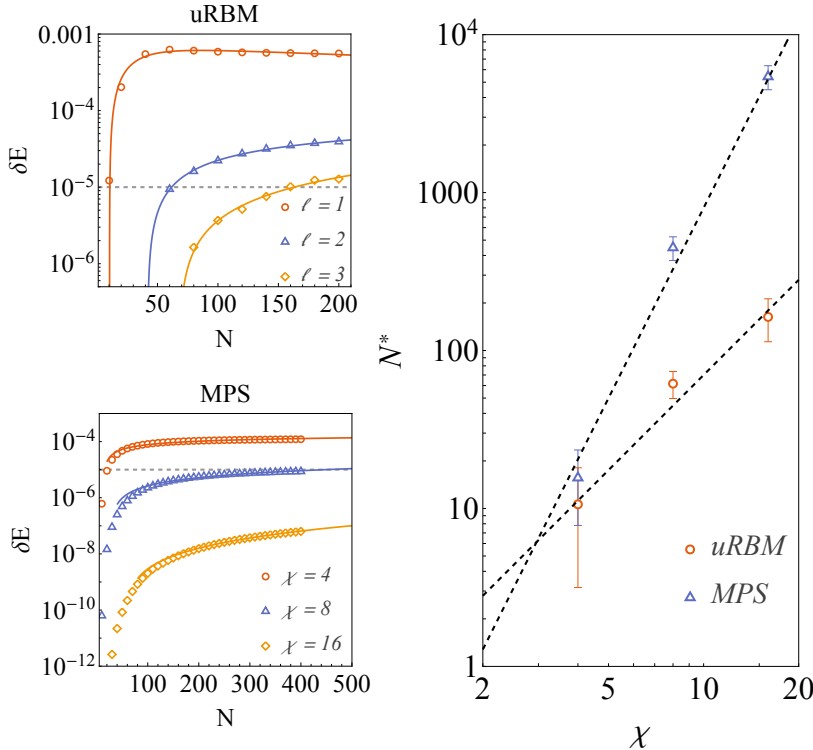

Figure 4: (left panels) Finite size scaling analysis of the relative error in the ground-state energy of the critical Ising chain for different many-body wave function ansatz (uRBM and MPS); the grey dashed lines correspond to the accuracy goal $10^{-5}$. (right panel) Scaling of the largest system size $N^*$ which can be described with an energy accuracy $\leq 10^{-5}$ for the two different ansätze. The black dashed lines are power-law fits as reported in the main text.

tations becoming better by increasing the number of variational parameters. Therefore, we investigate this aspect at the critical point (the more computational demanding case), where we perform a scaling analysis of the accuracy in the energy estimation for the different wave function representations. It turns out that, for equal hidden variable density $\alpha = \ell$, the uRBM overtakes the RBM; however, the canonical MPS with the same bond dimension $\chi = 2^{\ell+1}$ of the uRBM remains highly more accurate than any NN representation (see Fig. 3 right panel). For example, for $\alpha = \ell = 2$, we obtain $\delta E_{RBM} \simeq 0.8 \cdot 10^{-4}$, $\delta E_{uRBM} \simeq 0.16 \cdot 10^{-4}$, whilst the MPS with $\chi = 2^{\ell+1} = 8$ gives $\delta E_{MPS} \simeq 0.15 \cdot 10^{-5}$.

Once established that the uRBM with the Ising-like geometry gives better estimates of the ground-state Ising energy with respect to the RBM variational wave function, we may now concentrate on a more systematic Finite Size Scaling (FSS) comparison between uRBM and MPS. Indeed, in order to infer about a possible definition of the *descriptive power* of a given many-body wave-function ansatz, we decided to proceed in the following way: *(i)* We fixed the level of accuracy to be $10^{-5}$ so as to have a good interpolation of the uRBM data; *(ii)* we extract the largest system size $N^*$ whose ground-state energy can be estimated within that accuracy goal; *(iii)* we analyse the behaviour of $N^*$ as a function the (effective) bond dimension $\chi$, which indeed gives the algorithmic complexity of the energy minimisation procedure for both ansätze. Once again, we concentrate the FSS analysis to the more computational demanding critical chain (i.e. $\lambda = 1$). We perform numerical simulations with sizes $N \in [10, 200]$ for the uRBM with $\ell \in \{1, 2, 3\}$, and $N \in [10, 400]$ for the MPS with $\chi \in \{4, 8, 16\}$.

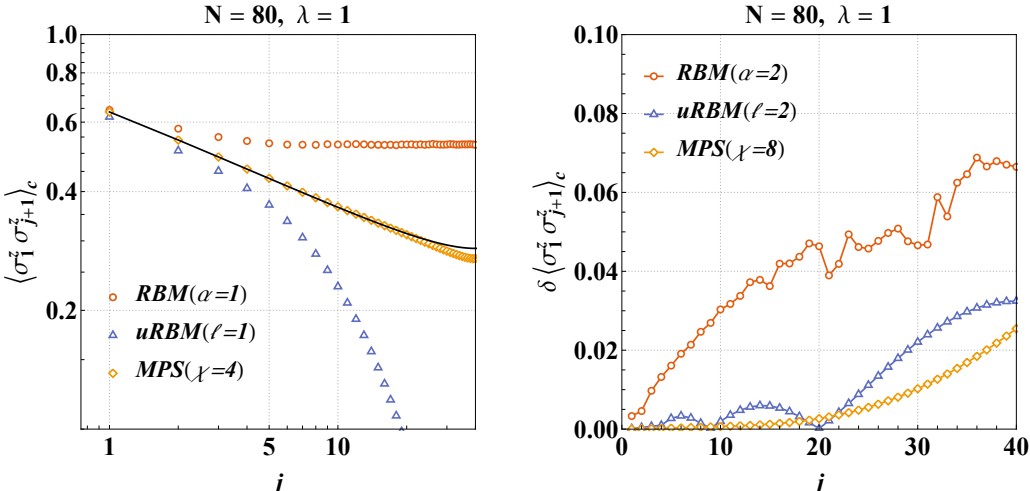

Figure 5: (**left**) Two-point connected correlation function in log-log scale at the critical point for different variational ansatz and smaller description, i.e. $\alpha = \ell = 1$ and $\chi = 2^{\ell+1} = 4$. Black full line are the exact analytical results. (**right**) Relative error from the exact data when larger NN representations are considered; here we compare RBM/uRBM with $\alpha = \ell = 2$ with the canonical MPS with $\chi = 2^{\ell+1} = 8$. All the data for RBM has been obtained by using the optimised wave functions in Ref. [13].

In the left panels of Fig. 4 we report the data (symbols) together with the best power-law fits $\delta E = a + bN^c$ (full lines). From the best-fit parameters and their errors we obtain the estimate of $N^*$, where the effective bond-dimension of the uRBM is $\chi = 2^{\ell+1}$. A different scaling of the *descriptive power* of the two ansätze is reasonably clear from the right panel of Fig. 4: The black dashed lines indeed show a fair agreement with $N^* \sim \chi^{4.2}$ for the MPS variational wave-function, while only $N^* \sim \chi^{1.9}$ for the uRBM one.

Let us stress once more that, the performances of the uRBM to characterise the low-energy properties of the Ising chain are strictly related to the fact that such variational ansatz somehow encode the Ising geometry. In principle, we do not expect the same degree of accuracy when analysing different 1D critical models; this is what happen, for example, when trying to characterise the ground-state energy of the XXZ spin-1/2 chain in the gapless phase, where the performances of both the uRBM and the MPS are one/two orders of magnitude worst.

Even though the different variational ansätze may give reasonable estimates of the ground-state energy, it is worth investigating the large-distance behaviour of correlation functions. Indeed, at the critical point, we expect a power-law decay of the two-point connected correlation function $\langle \sigma_1^z \sigma_{j+1}^z \rangle_c = \langle \sigma_1^z \sigma_{j+1}^z \rangle - \langle \sigma_1^z \rangle \langle \sigma_{j+1}^z \rangle$, as far as $j \ll N$. However, the MPS structure of the variational ansatz introduces an unavoidable fictitious correlation length. Moreover, the fact that the uRBM energy estimate is better than the RBM estimate (see Ref. [13, 50] for a comparison), implies that the uRBM may give a better estimate at the level of the correlation functions as well. With this respect, in the left panel of Fig. 5, we compare the connected two-point function $\langle \sigma_1^z \sigma_{j+1}^z \rangle_c$ evaluated in the optimised uRBM with $\ell = 1$ against the same two-point function evaluated in the unconstrained MPS with auxiliary dimension $\chi = 4$. In order to have the same number of hidden variables in both NN representations, we show the RBM correlations with $\alpha = 1$, which have been obtained by sampling the optimised wave-function in Ref. [13] over $10^6$ configurations. We focus our analysis to the critical point, where a larger deviation from the exact data is expected. From the figure it is clear that, the canonical

MPS is largely better than the neural-network representation.

In a way, the RBM suffers from a sort of over-estimation of the long-range correlations due to the presence of unphysical long-range couplings between hidden and physical variables; on the contrary, the over-constrained structure of the coMPS representation of the uRBM reflects into a stronger exponential decay of the two-point correlations. However, there is the possibility for those functions obtained by optimised NNs to be improved by the inclusion of further layers in the ansatz, which can increase the degree of correlations.

Indeed, when the number of hidden variables is increased to $\alpha = \ell = 2$ in such a way to reach the same energy accuracy of the MPS with $\chi = 4$ (see Fig. 3, right panel), the corresponding NN description of the correlation function improves as well and reaches that of the MPS with $\chi = 4$. However, it still remains less accurate with respect to the MPS representation with an auxiliary dimension $\chi = 2^{\ell+1} = 8$ (see right panel in Fig. 5). In particular, for distances $j \lesssim 20$ lattice sites, the RBM relative error remains $\lesssim 5\%$, the uRBM gets an error $\lesssim 0.6\%$ whilst finally the MPS reaches a better accuracy with an error $\lesssim 0.16\%$.

From this point of view a structured neural network states, namely a uRBM, seem better tailored than an RBM, at least, to deal with short-range one-dimensional systems. Once again, we may stress that when simulating with a uRBM, the coMPS representation should be used to calculate energy and further expectation values for higher accuracy and lower computational time.

## 3.2 The 2D Heisenberg model

We now extend our analysis to two-dimensional systems as well, by considering the 2D Heisenberg antiferromagnetic model, whose Hamiltonian is

$$H_H = \sum_{i,j=1}^{N} \sum_{\gamma} \left( \hat{\sigma}_{i,j}^{\gamma} \hat{\sigma}_{i,j+1}^{\gamma} + \hat{\sigma}_{i,j}^{\gamma} \hat{\sigma}_{i+1,j}^{\gamma} \right), \tag{14}$$

where $\gamma$ runs over $\{x, y, z\}$. We assume here periodic boundary conditions. The ground state of Heisenberg Hamiltonian is characterised by power-law decaying correlations, thus being a perfect two-dimensional benchmark.

As already stressed, an RBM is characterised by an exponentially large bond dimension and seems to work pretty well in the presence of long-range interactions and correlations [47]. Here, in order to avoid the expensive optimisation procedure for a PEPS wave function, we only consider the state-of-the-art RBM variational results in Ref. [13] and compare its descriptive power against a Tree Tensor Network (TTN) [34, 54–57] representation for the 2D Heisenberg ground-state. A TTN is a loop-free Tensor Network which can be efficiently contracted and it is characterised by a finite bond dimension $\chi$ which enforces the maximum amount of entanglement for any bipartition of the 2D state to be finite.

In our TTN simulations, we consider both $8 \times 8$ and $10 \times 10$ sizes; the former is more suitable for a TTN algorithm since it can fully exploit the binary-tree structure. We compare the estimated energy density with the best known results obtained via Quantum Monte Carlo [53] and show the relative deviation in Fig. 6 for both system sizes. In both cases (for $L = 8$ and $L = 10$) the errors of the Quantum Monte Carlo results are below $10^{-5}$ and therefore negligible compared to both, the TTN as well as the RBM.

The $10 \times 10$ case presents a geometry which is much less easy to adapt for TTN computations; nevertheless we are able to reach the same accuracy of an RBM with $\alpha = 1$ by only keeping $\chi = 340$ states (see Fig. 6), which should be compared with the RBM equivalent bond dimension $\chi_{RBM} \sim 2^{\alpha N^2} = 2^{100}$. Let us point out that in the $8 \times 8$ case, when a better-suited TTN geometry can be used, we are able to reach one order of magnitude better precision. With a relatively small bond dimension $\chi = 700$ we already almost meet the RBM results with

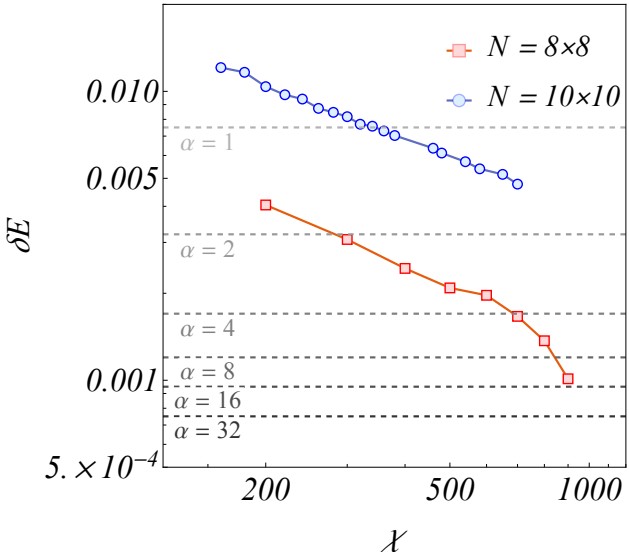

Figure 6: Relative error of the 2D Heisenberg ground-state energy compared with the best available estimates obtained by the finite-size Quantum Monte Carlo analysis in Ref. [53]. Symbols are TTN results for two different system sizes as function of the maximum bond dimension. Dashed lines represent RBM accuracy from Ref. [13] for the $10 \times 10$ system and different hidden variable densities $\alpha \in [1, 32]$.

$\alpha = 16$ (whose bond-dimension scale as $\chi_{RBM} \sim 2^{16.64}$). Let us stress that, this huge difference in the bond dimension scaling, suggests that such parameter is not a sensible measure of complexity for a Neural Network.

At this point, we may wonder how well different representations reproduce two-point correlation functions. Due to the $SU(2)$ symmetry of the Heisenberg Hamiltonian, we expect all correlations $\langle \hat{\sigma}_i^{\gamma} \hat{\sigma}_j^{\gamma} \rangle_c$ being independent of $\gamma$ when evaluated in the exact ground state. In the TTN framework, we enforced $U(1)$ symmetry along the $\hat{z}$ axis which provides $\langle \hat{\sigma}_j^{x,y} \rangle = 0$ thus the connected correlations are more accurate in the $\hat{x}$-$\hat{y}$ plane; we therefore compute correlations along the $\hat{x}$ axis. In the RBM case, we considered correlations in the $\hat{z}$ axis, since by construction they are more accurate and easier to measure here; again in this case, the RBM correlations have been obtained by sapling over $10^6$ configurations the optimised wave function in Ref. [13]. With this prescription we are sure to compare the best estimates in both representations.

In Fig. 7 we show the TTN correlations for the $10 \times 10$ size and different bond dimensions, and compare them to the RBM with $\alpha = 1$ and 2. It is clear that correlations are growing and getting better with an increasing number of variational parameters. However, the insight from this comparison is twofold: (i) the exponentially large bond-dimension of the RBM is not a guarantee for this representation to be able to encode power-law correlations in critical 2D short-range interacting models and thus to overtake a Tensor Network representation based on finite bond-dimension; (ii) when both TTN and RBM get the same accuracy in the energy (i.e. $\alpha = 1$ and $\chi = 340$ for the $10 \times 10$ size), TTN is more accurate for characterising the correlation functions. However, regarding the latter, we mention that this finding and especially the magnitude of the difference in characterising the correlation functions may as well be dependent on the model of investigation. Thus, the generalisation of this statement has to be investigated in future work.

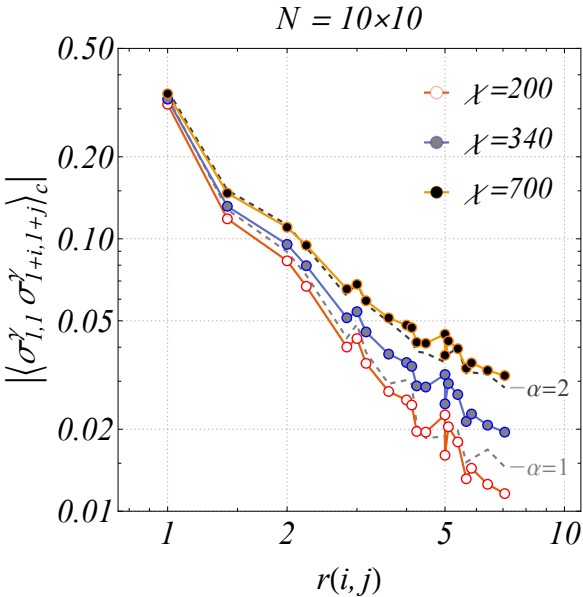

Figure 7: Connected correlation function in the TTN representation of the ground state of the $10 \times 10$ Heisenberg Hamiltonian for different bond dimensions (symbols) vs the distance $r(i, j) \equiv [i^2 + j^2]^{1/2}$, where $i \leq j \in \{0, 5\} \times \{0, 5\}$. The dashed grey lines represent correlations obtained by sampling the RBM optimised wave function in Ref. [13].

Let us further point out that in this particular benchmark we may as well exploit the $SU(2)$ symmetry in the TTN simulations thus allowing us to: (1) dramatically increase the accuracy in the estimated energy; (2) reduce the effective bond dimension and thereby the computational time since for non-abelian symmetry we may only work within the symmetry multiplet spaces; (3) drastically improve the connected correlations since we enforce $\langle \hat{\sigma}_j^\gamma \rangle = 0$, as well as $\langle \hat{\sigma}_i^\gamma \hat{\sigma}_j^\gamma \rangle_c$ to be equivalent independent on $\gamma$. While (3) can be achieved as well for the RBM when encoding symmetries, (2) does not apply for RBMs. Thus, for RBMs, as for the TTN, we would expect to gain a higher precision in the final results when exploiting the $SU(2)$ symmetry as a result of (3). However, we would not expect such a dramatic increase of the computational time as in the case of the TTN, which in return enables the TTN to achieve higher bond dimensions and thereby to further increase the accuracy additionally to point (3).

## 4 Discussions

We investigated the efficiency of Neural Network quantum states with respect to Tensor Network quantum states when used to describe the ground-state of critical short-range interacting Hamiltonians both in one and two dimensions.

We pointed out and exploited the "constrained" Tensor Network State (coMPS/coPEPS in 1D/2D) representation of the neural-network wave function. As a matter of fact, RBM and uRBM have very different representations. Even though the coMPS associated to the RBM has an auxiliary dimension which scales exponentially with the number of hidden variables, it still struggles to properly describe the ground-state correlation functions of critical Hamiltonians.

Indeed, in the 1D Ising case, a much smaller coMPS dimension associated to a much more constrained uRBM wave function gives a more accurate description if compared to the RBM

parametrisation. However, it turns out that, for equal auxiliary dimension, standard MPS algorithms give a more acurate description of the many-body ground state. In addition, since the performances of bothe, the uRBM (in the coMPS representation) and the MPS algorithm, scale with the (effective) auxiliary dimension $\chi$, an accurate FSS analysis suggests a possible definition of the *descriptive power* of a variational wave-function: namely, the largest system size which can be faithfully (i.e. within an accuracy goal) described by a given ansatz. As a matter of fact, the descriptive power of the un-constrained MPS outperforms the uRBM descriptive power, when both are used to approximate the ground-state of a critical Ising chain.

In this sense, the exponentially large auxiliary dimension of the coMPS associated to a generic RBM seems not enough to provide a good characterisation of the long-range correlations in the critical Ising quantum chain. In order to obtain more accurate estimates, system-dependent deep neural network states, namely a uRBM with $\ell \gg 1$, have to be properly optimised. Our explicit coMPS representation of the uRMB variational ansatz can be eventually combined with Monte Carlo techniques thus overcoming the limitation, pointed out in Ref. [50], of sampling over the hidden-variable configurations; thus making the Monte Carlo approach also effective for Hamiltonians where the sign-problem occurs. Moreover, an optimised uRBM can be used to optimally initialise convolutional NN algorithms, so as to speed up the computations.

In 2D we compared the RBM representation against the TTN representation when both are used to approximate the ground-state many-body wave function of the two-dimensional Heisenberg Hamiltonian. As expected, both methods are well suited to describe the 2D many-body quantum system. However, when reaching the same level of accuracy in the energy estimate, a TTN is more precise in characterising long-range correlations, even though they are employing a strictly finite bond dimension far below the mapped counterpart of the RBM representation. This, from the one side, leaves no doubt that the exponentially large auxiliary dimension of the RBM does not ensure an adequate descriptive power; from the other side, it leaves us with the open question on how to properly estimate the information which is encoded in a NN, so as to define a proper *measure of complexity* for a neural network quantum state.

## 5 Acknowledgments

We are very grateful to Giuseppe Carleo for valuable comments which allow to improve the manuscript.

**Funding information.** — This work was supported by the BMBF and EU-Quantera via QT-FLAG and QuantHEP, the Quantum Flagship via PASQuanS, the DFG via the TWITTER project, the Italian PRIN2017 and by the BIRD2016 project 164754 of the University of Padova.

**Author contributions.** — All authors agreed on the approach to pursue, and contributed to the interpretation of the results and to the writing of the manuscript. L. D. conceived the work. M. C. and T. F. performed the numerical simulations. S. M. supervised and merged the research. The authors declare no competing interests.

## A  1D numerical simulations

The numerical simulations for the ground-state optimisation in the Ising quantum chain have been performed by means of different approaches. In particular, the optimisation of the canonical MPS has been done by using the well established DMRG algorithm [25]. In our algorithm,

we fixed the auxiliary dimension $\chi$ to remain constant. A preliminary "infinite" size procedure enlarges the system up to the desired linear dimension $N = 80$. Thereafter, the usual "sweeps" procedure locally optimises the MPS wave function. The algorithm is stopped when the energy difference between two consecutive sweeps is less that the machine precision.

In the uRBM approach, we exploited the coMPS representation of the ansatz for the wave-function so as to get very accurate results. If $\mathbf{M}^\sigma$ is the local tensor depending on $2\ell + 1$ real variational parameters $\vec{K}$, we introduced the local operator-dependent transfer-matrix

$$\mathbf{T}_{\hat{O}} = \sum_{\sigma,\sigma'} \langle \sigma' | \hat{O} | \sigma \rangle \, (\mathbf{M}^*)^{\sigma'} \otimes \mathbf{M}^\sigma, \tag{15}$$

which implicitly depends on the variational parameters $\vec{K}$, and globally minimised the energy density of the Ising quantum chain

$$\varepsilon[\vec{K}] = -\frac{\mathrm{Tr}[\mathbf{T}_{\hat{\sigma}^z} \mathbf{T}_{\hat{\sigma}^z} \mathbf{T}_{\hat{I}}^{N-2}] + \lambda \,\mathrm{Tr}[\mathbf{T}_{\hat{\sigma}^x} \mathbf{T}_{\hat{I}}^{N-1}]}{\mathrm{Tr}[\mathbf{T}_{\hat{I}}^N]}. \tag{16}$$

In the simplest case of $\ell = 1$ we used the Mathematica builtin routine `NMinimize` which turns out to be stable and it efficiently converges to the global minimum. However, for larger parameter spaces, namely $\ell = 2$ and $3$, the Mathematica routine does not give the expected improvement in the energy minimisation, and it gets stacked on some local minimum. We thus improved the global minimisation by randomly reducing the dimension of the parameter space wherein `NMinimize` has to look for a global minimum. In practice, we proceed in the following way:

1. We randomly initialise a real vector $\vec{K}$ which contains the $2\ell + 1$ variational parameters.

2. We randomly construct a $(2\ell+1) \times (2\ell+1)$ orthogonal matrix $\mathbb{R}$ and define the variational parameter vector in the new basis $\vec{K}' = \mathbb{R}^T \vec{K}$.

3. We pick up three components of the vector $\vec{K}'$ and promote them as variational variables, thus defining the 3-variable dependent vector $\vec{K}'(x, y, z)$ where $\{x, y, z\}$ is a three dimensional subset of variational parameters. We thus transform back the vector to the original basis so as to have $\vec{K}(x, y, z) = \mathbb{R}\vec{K}'(x, y, z)$. We minimise $\varepsilon[\vec{K}(x, y, z)]$ with respect to $\{x, y, z\}$ by using `NMinimize`, thus finding the best parameters $\{x^*, y^*, z^*\}$. If the new optimised energy density is lower than the actual best estimate, we upgrade the solution $\vec{K} = \vec{K}(x^*, y^*, z^*)$.

4. We repeat point 3 for all possible different way of taking three components of the vector $\vec{K}'$, i.e. $\binom{2\ell+1}{3}$. Thereafter, we go back to point 2 and repeat the procedure.

5. We stop the recipe when the difference in the two best energy estimates is less than $10^{-9}$.

# B  2D Numerical simulations

The simulations for the ground-state computation of the isotropic 2D Heisenberg model have been done using a binary Tree Tensor Networks (TTN). In this approach each tensor within the Network combines two sites to one coarse-grained virtual site (or bond link), resulting in the hierarchical tree structure. The optimisation of the TTN, as well as the calculation of the observables for the optimised ground-state, were obtained following the description for loopless Networks in Ref. [34]. For all simulation the $U(1)$ symmetry has been exploited. Furthermore,

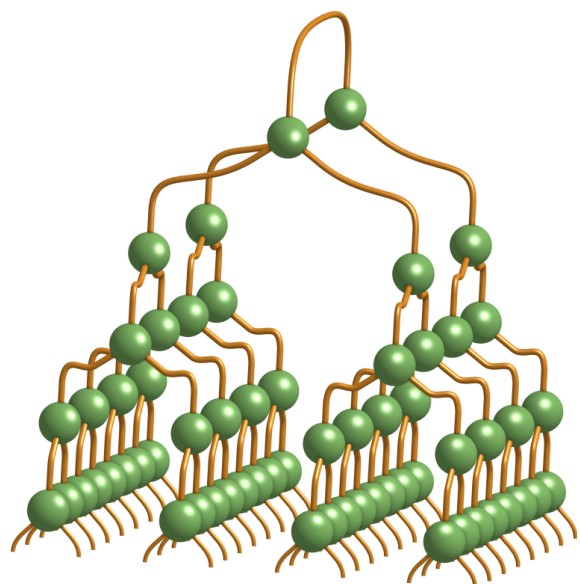

Figure 8: Binary TTN structure for the $8 \times 8$ simulations. The tensors (green) each merges two sites of the lower layer to one bond link (brown). The mapping here groups alternatingly in $x$- and $y$-direction from layer to layer starting from the physical sites of the $8 \times 8$ lattice.

for each simulation the Network was randomly initialised within the zero-magnetisation symmetry sector and within the given bond dimension $\chi$.

For the $10 \times 10$ simulations, the physical sites $j \in \{1, \ldots, 100\}$ of the TTN were assigned in a zig-zag pattern to the two-dimensional lattice, such that the lattice site $(x, y)$ (with $x, y \in \{1, \ldots, 10\}$) is mapped to the TTN site $j = x + 10 \cdot (y - 1)$. Thereby, the system is coarse-grains in $x$-direction first at the lower layers of the tree, and afterwards at the upper layers in $y$-direction. Thus the simulation is biased towards the $x$-direction as the topology of the TTN is not well suited to capture correlations in $y$-directions. This makes the $10 \times 10$ system size in general not ideal for a TTN approach.

In the case of the $8 \times 8$ system size, the TTN was arranged, such that the grouping within the network is done in an alternating form from layer to layer, as depicted in Fig. 8. Thus the Tensors in the TTN is coarse graining the system in local plaquettes and thereby better capture the correlations within these plaquettes. This mapping leads to a more precise description, which can be observed in the energy being an order of magnitude more accurate (see Fig. 6).

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
