# Peer review of "On the descriptive power of Neural-Networks as constrained Tensor Networks with exponentially large bond dimension"

_SciPost Physics Core, doi:SciPost Phys. Core 4, 001 (2021)_

## Round 3 · Referee Report · Anonymous (Referee 1) · 2020-6-24

Strengths

1) Systematic comparison of the accuracy of the coMPS representation of (unrestricted) Boltzmann machines against canonical MPS for prototypical condensed matter physics models.

Weaknesses

1) Use of an unfavorable metric and parameters (w.r.t. the neural networks) for the comparison. 2) Results are based on very specific method to optimize the variational parameters, without checking alternative methods/minimizers. 3) The authors compare the optimization of the coMPS representation of the (u)RBM, rather than optimizing the parameters of the (u)RBM itself, yet they compare against literature results from an RBM.

Report

The authors aim to present a systematic comparison constraint MPS (coMPS) representations of (u)RBM neural networks (NN) with canonical MPS as ansatz for quantum many body wave functions of prototypical condensed matter physics models. I consider this a valuable objective in the booming field of applications of neural networks in quantum many body physics.

Their main metric for comparison is the effective bond dimension of the coMPS representation for the uRBM rather than the number of d.o.f. of the uRBM itself, which is a significantly different number of parameters to be optimized. Despite the very different representations the authors compare results from RMBs, coMPS and canonical MPS simulations on the basis of the bond dimension. I do not consider this metric appropriate when RBMs are included. Comparing coMPS and MPS bond dimensions only is certainly better, yet still even then it is unclear which of those two can be optimized more/less easily in practice.

Optimization results are highly dependent on the ansatz/representation AND the methods employed to optimize the auxiliary degrees of freedom. I am not aware though, of a systematic comparison between the accuracy that can be practically reached within the coMPS representation and the uRBM itself, nor if coMPS and MPS can be optimized similarly well. This manuscript certainly does not provide this information. The different representations have different constraints, such that optimization in one of them might be less prone to local minima than the others. While the authors employ a randomized procedure to improve the minimum search with NMinimize they do not consider different minimization algorithms. A different choice of solver can also significantly alter the outcome.

Concerning the points above the authors need reformulate the manuscript in such a way that they clearly and transparently state from the begining what they actually compare, their methodology and its limitations.

In Sec. 3.1 the authors mention that algorithms based on the coMPS representation are more efficient in terms of CPU time as they are based on matrix-matrix multiplications. To my knowledge variational Monte Carlo is based on matrix-vector multiplications and following even more efficient? I think a direct comparison of the 'time-to-solution' is more appropriate here.

In Sec. 3.2 the authors compare results from the TTN of finite size clusters against the expectations values of the AFM Heisenberg model in the thermodynamic limit. The displayed errors are thus not the relative errors with respect to the accuracy of solving a certain finite size system. Since the reference values are obtained from sampling, errorbars should be provided or at least mentioned, e.g., if they are smaller than the symbol size. In Fig. 6 the legend key reads N = 8, 10, but probably should L = 8, 10 where L is the linear dimension such that N = L^2?

The stated lesson that the bond dimension of the RBM (should be coMPS) is not guaranteed to be able to encode power law correlations, is not substantiated by the very limited range of \alpha=1,2 presented. The lesson that for the same energy accuracy the TTN is more accurate, cannot be generalized from the very limited results.

On the point raised that accounting for additional symmetries would allow to dramatically increase the TTN accuracy: Additional symmetry constraints can also be implemented in the method of Ref. 13.

The authors mention twice that the coMPS representation should be used to evaluate expectation values for higher accuracy and lower computational time. I would stress though, that (to my knowledge) the evaluation of expectation values even with stochastic methods is not the problem such that, while maybe computationally more expensive, arbitrary precision can be reached. The challenge and computational cost lies in the optimization of the wave function. In this regards I suggest to rephrase the authors statement on p. 7, last paragraph.

I do not consider the manuscript in its current form suitable for the high standards of SciPost Physics. If the editor decides to advance the manuscript after revision, I think it may be more suited for SciPost Physics Core.

---

## Round 3 · Referee Report · Anonymous (Referee 2) · 2020-7-15

Strengths

1- Comparison of a variety of variational states using various extrapolation schemes 2- Mapping between RBM and co-mps

Weaknesses

1- The comparison is done between states with very different number of variational parameters 2- Different methods were used for optimising the MPS states compared to the neural network states. This makes it difficult to compare 'descriptive power', which the authors attempt to do. 3- Slightly misleading title/abstract/conclusion considering only a small class of neural network states (RBM and uRBM) and tensor network states were investigated.

Report

The authors presents a class of neural network states (RBM/uRBM) as constrained tensor network states and attempts to make a comparison with standard unconstrained tensor networks, in terms of 'descriptive power'. It is an interesting work considering the increasing popularity of such variational states and the lack of a detailed comparison of the expressiveness of the various ansatz.

However, it is not easy to evaluate the quality of their comparison for two main reasons: 1) the states being compared have very different number of free variational parameters 2) the different states are optimised with different methods, making difficult to disentangle the expressiveness of the ansatz from the effectiveness of the optimisation.

Moreover, considering that only very narrow class of neural networks (RBM/uRBM) and tensor networks (MPS/TTN) have been considered, it would perhaps be prudent adjust the title/abstract/conclusion to more accurately reflect scope of the paper.

Despite the above points, I think the paper do highlight some important aspects when attempting to compare different variational ansatz. While I do not think the paper should be published in SciPost Physics, I think with some revision, the paper could be published in SciPost Physics Core.

---

## Round 4 · Referee Report · Anonymous (Referee 2) · 2021-1-13

Strengths

1- honest evaluation of the expressiveness of different variational states together with their appropriate optimization approach

2- Interesting mapping from

Weaknesses

2- the use of different optimization approaches entangles expressiveness with optimizability

Report

The authors have sufficiently addressed the concerns I raised.

There are not many works seriously comparing the descriptive power of neural networks with other variational ansatz. Even though the use of different optimization approaches makes the comparison slightly more obscure,
I still do believe that the work deserves a publication in SciPost Physics Core.

---

## Round 4 · Author Response

REPLY TO REFEREE 1
WEAKNESS:
We thank the referee for pointing out these weaknesses when comparing different variational methods. However, one central message of the manuscript is indeed making the scientific community aware of the difficulties arising when naively comparing these ansätze. In particular, many references in this scientific community compare the neural network states to exponential large Tensor networks without performing fundamental analysis of the actual descriptive power of those methods. Thus, with our manuscript, we address the lack of a detailed comparison of the expressiveness of these increasingly popular, variational ansätze. Therefore, we are very happy and see it as a strong point of our manuscript that the referee successfully pointed out these difficulties in comparing the representations which can start a proper discussion of a more detailed comparison.

In particular, for point 1) we stress that often the number of variational parameters is far from being a proper indicator for the accuracy in a proper comparison. For instance, in the AKLT model, the MPS is exact with bond dimension 2. However, the same model would require $O(N^2)$ hidden units in the RBM. On the other hand, an RBM is well suited for representing the Laughlin wave function with $N(N-1)/2$ hidden units in contrast to the here actually required exponentially large bond dimension for tensor networks. Thus, concluding, we are not convinced that fixing the number of variational parameters in general leads to a fair comparison.
For point 2), we point out that the different optimisation techniques used in the comparison are state-of-the-art for each of the variational ansätze. Thus, for each ansatz, the chosen optimisation technique delivers state-of-the-art results. We are convinced that a fair comparison should be exactly this: Comparing the best possible outcomes of both approaches.

Given that we explicitly state the procedure on how we perform the analysis, we are convinced that the results (i) can be used for a proper comparison and (ii) most certainly can help to encourage a scientific discussion about the differences of the methods and thereby inspire novel insights on the yet unknown connections between the approaches.

For point 3), We thank the referee for pointing out the slightly misleading parts in the manuscript. For the sake of clarity, we changed critical parts in the abstract, introduction and minor parts in the conclusion.

REPORT:
Point 1) and 2): We thank the referee for pointing out these issues and refer to the answer above.

Moreover, We thank the referee for pointing out the slightly misleading parts in the manuscript. For the sake of clarity, we changed critical parts in the abstract, introduction, main text and minor parts in the conclusion, see the revised version of the manuscript.

Finally, We thank the referee for pointing the scientific importance of our manuscript and would like to inform that we performed a proper revision based on the valuable comments and suggestions of both referees. As both referees suggested, we will transfer our manuscript for publication in SciPost Physics Core.

REPLY TO REFEREE 2
We thank the referee for pointing this strength of our manuscript and would like to point out that the mentioned comparison is not only performed in 1D with an MPS against NNs, but as a further strong point of our manuscript in 2D with a TTN against NNs as well.

REPORT:
We thank the referee for pointing out the issue when comparing different variational methods. However, we point out that often the number of variational parameters is far from being a good indicator for the accuracy in a proper comparison. For instance, in the AKLT model, the MPS is exact with bond dimension 2. However, the same model would require O(N^2) hidden units in the RBM. On the other hand, an RBM is well suited for representing the Laughlin wave function with N(N-1)/2 hidden units in contrast to the here actually required exponentially large bond dimension for tensor networks. Thus, concluding, we are not convinced that fixing the number of variational parameters, in general, is a completely appropriate metric either for a fair comparison. Thus, finding a completely fair way of comparing the methods remains an open question as we point out in our manuscript.

Given that we explicitly state the procedure on how we perform the analysis, we are convinced that the results (i) can be used for a proper comparison and (ii) most certainly can help to encourage a scientific discussion about the differences of the methods and thereby inspire novel insights on the yet unknown connections between the approaches.

While we agree with the statement that a different choice of solver might in principle alter the outcome, we point out that the different optimisation techniques used in the comparison are state-of-the-art for each of the variational ansätze. Thus, for each ansatz, the chosen optimisation technique delivers state-of-the-art results. We are convinced that a fair comparison should be exactly this: Comparing the best possible outcomes of both approaches. Moreover, the referee’s critique is possible for any numerical minimization result, however, this does not prevent the scientific community to build new physics on those possibly flawed results.

We thank the referee for pointing out the parts to be addressed for more clarity and transparency. Thus, for the sake of clarity, we changed critical parts in the abstract, introduction, main text and minor parts in the conclusion. We are confident that our manuscript has been improved and is now more transparent in our comparison and its limitations. Thus, we thank the referees for their support in improving our manuscript.

The formulation used to describe the CPU time was misleading for the reader. The main message was that the coMPS is more efficient in calculating the expectation values (which is a part of the optimisation). The fact that they are based on matrix-matrix-multiplications was independent of the above statement and used to clarify that the required resources scale with O(\chi^3).
However, we realized that both massages are already mentioned and we chose to remove them to reduce redundancy.

We do not compare the expectations values of the AFM Heisenberg model in the thermodynamic limit. However, we compare our results with Monte Carlo simulations of the exact same finite size systems (for L=8 and L=10) with the same boundary conditions. The paper, in which these results were published aims to calculate the expectations values of the AFM Heisenberg model in the thermodynamic limit by extrapolating the results for finite system sizes L={4,6,8,10,12,14,16}. In both cases, for L= 8 and L= 10, the errors of the Quantum Monte Carlo results are below 1E−5 and therefore negligible compared to both, the TTN as well as the RBM, which we now state in our manuscript.

We changed the figure 6 accordingly and thank the referee for his/her comment.

We improved the wording for the comparison between RBM and TTN, i.e., we now explicitly state that our findings apply to the models we analysed, but that the generalisation of our statements remains an open question, as the outcome for the different method might be highly model specific.

The referee is correct, that additional symmetry constrains can also be implemented in the method of Ref. 13. However, while encoding symmetries in RBMs will indeed improve the connected correlations since they enforce $\langle\sigma^\gamma_j\rangle= 0$, as well as $\ langle\sigma^\gamma_i \sigma^\gamma_j\ rangle $ to be equivalent independent on $\gamma$, the actual computational gain is higher in the case of a TTN. Thus, for RBMs, we would expect to gain a higher precision in the final results when exploiting the $SU(2)$ symmetry as a result of the improved correlations. However, we would not expect a dramatic increase in the computational time in contrast to the case of the TTN. Here we explicitly reduce the effective bond dimension by changing the basis to only work within the symmetry multiplet space. This drastically decreases the computational resources required which in return enables the TTN to achieve higher bond dimensions and thereby to further increase the accuracy additionally to the gains resulting from the enforced expectation values.
We added a paragraph to clarify this benefit as indeed this message might not have been clear in our prior version of the manuscript.

We added a statement on page 7 to clarify that the coMPS approach drastically improves the evaluation of expectation values only and not the optimisation itself.

Finally, We thank the referee for pointing the scientific importance of our manuscript: we performed a proper revision based on the valuable comments and suggestions of both referees. In conclusion, following the referees suggestion, we are going to submit our manuscript for publication in SciPost Physics Core.

---

## Editorial Decision

published